# Parallelizable Microfluidic Platform to Model and Assess In Vitro Cellular Barriers: Technology and Application to Study the Interaction of 3D Tumor Spheroids with Cellular Barriers

**DOI:** 10.3390/bios11090314

**Published:** 2021-09-03

**Authors:** Arya Lekshmi Nair, Lena Mesch, Ingo Schulz, Holger Becker, Julia Raible, Heiko Kiessling, Simon Werner, Ulrich Rothbauer, Christian Schmees, Marius Busche, Sebastian Trennheuser, Gert Fricker, Martin Stelzle

**Affiliations:** 1NMI Natural and Medical Sciences Institute, University of Tübingen, Markwiesenstraße 55, 72770 Reutlingen, Germany; arya.lekshmi1604@gmail.com (A.L.N.); jschuette.tue@googlemail.com (J.R.); heiko.kiessling@live.de (H.K.); Simon.werner@nmi.de (S.W.); Ulrich.Rothbauer@nmi.de (U.R.); Christian.Schmees@nmi.de (C.S.); mariusbusche@hotmail.de (M.B.); 2Currently with Mimetas, De Limes 7, 2342 DH Oegstgeest, The Netherlands; 3Institut für Neuroanatomie und Entwicklungsbiologie, University of Tübingen, Österbergstraße 3, 72074 Tübingen, Germany; lena.mesch@uni-tuebingen.de; 4Microfluidic ChipShop GmbH, Stockholmer Str. 20, D-07747 Jena, Germany; Ingo.Schulz@microfluidic-chipshop.com (I.S.); hb@microfluidic-chipshop.com (H.B.); 5Institute of Pharmacy and Molecular Biotechnology, University of Heidelberg, Im Neuenheimer Feld 364, 69120 Heidelberg, Germany; s.trennheuser@uni-heidelberg.de (S.T.); gert.fricker@uni-hd.de (G.F.); 6Institute of Pharmacy, Pharmaceutical Biotechnology, Auf der Morgenstelle 8, D-72076 Tübingen, Germany

**Keywords:** cellular barriers, microfluidic device, TEER, tumor spheroids, Fourier-transform impedance spectroscopy

## Abstract

Endothelial and epithelial cellular barriers play a vital role in the selective transport of solutes and other molecules. The properties and function of these barriers are often affected in case of inflammation and disease. Modelling cellular barriers in vitro can greatly facilitate studies of inflammation, disease mechanisms and progression, and in addition, can be exploited for drug screening and discovery. Here, we report on a parallelizable microfluidic platform in a multiwell plate format with ten independent cell culture chambers to support the modelling of cellular barriers co-cultured with 3D tumor spheroids. The microfluidic platform was fabricated by microinjection molding. Electrodes integrated into the chip in combination with a FT-impedance measurement system enabled transepithelial/transendothelial electrical resistance (TEER) measurements to rapidly assess real-time barrier tightness. The fluidic layout supports the tubeless and parallelized operation of up to ten distinct cultures under continuous unidirectional flow/perfusion. The capabilities of the system were demonstrated with a co-culture of 3D tumor spheroids and cellular barriers showing the growth and interaction of HT29 spheroids with a cellular barrier of MDCK cells.

## 1. Introduction

Cells lining the vasculature form barriers separating blood from its surrounding tissues and play a vital role in the selective transport of solutes and other molecules across these barriers. Endothelial cells that constitute the lumen of intact blood vessels form a continuous monolayer that acts as a barrier between blood and the surrounding tissues [1,2]. Epithelial cells serve as a protective layer lining the inside and outside cavities of the human body. Both epithelial and endothelial cells are interconnected via specialized tight junctions or zona occludens that render selective permeability to these barriers [2]. Barrier integrity is crucial for physiological function and homeostasis. Barrier tightness varies between organs and also between different vascular segments of the same organ [1]. For instance, endothelial cells that make up the highly specialized blood–brain barrier (BBB) of the central nerve system (CNS) possess unique barrier properties and differ in their permeability compared to any other endothelial barrier of the body [3]. 

In many cases of chronic inflammation and disease, the integrity of the cellular barriers protecting vital organs of the human body tends to be disturbed. For instance, persistent neuroinflammation that has been reported in the pathology of almost all neurodegenerative disorders is accompanied by a disruption in the BBB integrity [4,5]. In addition, physical remodeling and disruption of endothelial cell barriers with increased permeability were observed during transmigration of tumor cells in the context of intravasation and extravasation, which are key steps associated with cancer metastasis [6,7,8,9]. Furthermore, hyperpermeability and loss of integrity of intestinal barriers were noted in the pathogenesis of several gastrointestinal disorders such as inflammatory bowel disease [10,11]. These examples highlight the importance of modelling cellular barriers to advance our understanding of barrier integrity. Significant progress has been made in clarifying the underlying mechanisms of endothelial and epithelial barrier function in both physiological and pathological contexts [12,13,14]. However, our understanding of barrier integrity and their disruption in events of trauma, injury or inflammation is far from complete. Therefore, in vitro models that better recapitulate the physiology, variability and complexity of cellular barriers are under rapid development and may be exploited to study inflammation, disease mechanisms and progression [15,16,17,18,19,20,21,22,23,24]. These models possess enormous potential in advancing drug development and discovery by developing techniques and approaches that could facilitate the delivery of therapeutic drug candidates across these barriers [25,26] to reach their target tissue and successfully treat diseases of organs protected by these barriers. 

To enable efficient drug screening and toxicity studies, in vitro models of cellular barriers must exhibit the characteristics and properties seen in vivo, such as the expression of junctional proteins and complexes. Transwell-based platforms are most often used to model cellular barriers in vitro; however, they lack critical properties such as the shear forces associated with physiological blood flow [27,28]. As a result, some cell types can lack important markers or transporters that are found in vivo, thereby limiting the suitability of these platforms to reflect their physiological relevance in vitro. Furthermore, these models lack the 3D microenvironment, complex cell–cell and cell–matrix interactions observed in vivo. On the other hand, animal models closely resemble the level of structural and functional complexity seen in humans, but they only allow limited control and manipulation of experimental conditions, raise ethical concerns and are also expensive [29], time consuming, have limited availability and are poor predictors of human outcomes due to species–specific differences [30,31]. Thus, there is a pressing need for improved in vitro systems that better mimic the tissue complexity and physiology [32,33]. 

With the advent of microfluidic organ-on-chip technology, it has become easier to model miniaturized healthy and diseased human tissues in vitro with increased physiological relevance. Advancement in microfabrication, microfluidics, biomaterials and tissue engineering techniques have been exploited to establish relevant test systems via the co-culture of multiple cell types, with cells often embedded in a hydrogel, to recapitulate the 3D spatial organization, complexity and heterogeneity of in vivo tissues [34,35,36]. A number of research groups have been successful in developing models of cellular barriers with important insights into cell culture conditions, morphology, viability and integrity of barriers in a microfluidic chip, and their potential in drug screening applications, particularly in the context of modelling the blood–brain barrier [18,20,21,23,37,38]. Also, devices to model aspects of cancer physiology and development have been presented [39,40]. However, most of these devices were fabricated from PDMS. While PDMS presents advantages such as ease of fabrication, biocompatibility and gas permeability, PDMS-based chips tend to adsorb substances [41,42], which is a major concern for drug screening applications, given that most of the drugs are hydrophobic in nature. In most cases, chip fabrication is based on soft lithography, which does not allow for seamless upscaling of device numbers. Yet, validation procedures and the demonstration of robustness and reproducibility in real world applications will clearly require the availability of a large number of devices. In addition, perfusion is often achieved by pressure-driven flow through tubing using syringe pumps. While tube-connected systems yield well-controllable flow rates, they are difficult to parallelize and are not easily adaptable to conventional cell culture and drug development workflows.

Therefore, we exploited microfluidic technology to model cellular barriers in a parallelizable microfluidic platform manufactured by microinjection molding of cyclic olefin copolymer (COC) comprising integrated electrodes to enable real-time transepithelial/transendothelial electrical resistance (TEER) measurement.

Continuous unidirectional flow was achieved by means of gravitation-driven flow in combination with an incubator compatible perfusion system. TEER was measured by employing a custom-made parallelized fast Fourier transform (FFT)-impedance spectrometer. As an example, and to demonstrate the versatility of this device, we established the complete workflow ranging from cell seeding to the generation of co-cultures of tumor cells growing in a hydrogel mimicking tumor extracellular matrix (ECM) along with a cellular barrier. Here, we show preliminary data to demonstrate how this system can be used to monitor the growth of micro-tumors and their interaction with a cellular barrier and migration into the perfusion channel.

## 2. Materials and Methods

### 2.1. Device Microfabrication

The microfluidic chip (MFC) was manufactured from cyclic olefin copolymer (COC) by microfluidic ChipShop GmbH, Jena, Germany. COC is transparent and exhibits low autofluorescence. It comprises ten micro channels with five different layouts of pillar size and gap width (Figure 1 and Figure A1, Table A1). The mold inserts used for injection molding were fabricated in-house at microfluidic ChipShop by ultraprecision micro-milling.

The chip body was manufactured by precision injection molding. The channel depth was 170 µm. The process yields devices with excellent optical properties and low substance adsorption [43]; the latter may be further reduced by functional coatings if necessary [44,45]. Shadow masks were made by laser machining in thin steel foils and used to deposit electrodes by vacuum evaporation of titanium (10 nm) acting as adhesion promoter and gold (100 nm). Shadow masks were aligned carefully with respect to the open microfluidic structures to generate electrodes on the side walls next to the cultivation areas and connecting leads (golden-colored structures in Figure 1) and were kept in place by means of permanent magnets located underneath the chip fixture.

Finally, microfluidic structures, electrodes and connecting leads were covered by bonding a thin (approx. 140 µm) COC foil onto the bottom of the chip, leaving connecting pads open to enable electric connection to the electrodes.

The described fabrication process is scalable in regard to numbers and yields devices in reproducible quality.

### 2.2. Integration of Gel Matrix

Each culture area is divided into two channels by a micro pillar array (Figure 1B): the abluminal side, which corresponds to tissue, is initially filled with a hydrogel solution comprising HT29 cells, which then polymerizes and establishes a smooth gel surface between two adjacent pillars (Figure 1C–E). Pillar spacing was designed in such a way (Figure A1) that adjacent pillars act as capillary stop valves that prevent the spillover of hydrogel solution into the luminal channel during priming. The luminal side, which represents the blood capillary, was filled with cell culture medium and was later seeded with cells, which form a cellular barrier on the gel interface, thus sealing the space between the pillars. Thus, the micro pillar array within each channel offered spatial control over the hydrogel that resembles the extracellular matrix in the chip. The reagents for the hydrogel were obtained from Cellendes GmbH, Reutlingen, Germany and mixed as indicated by the manufacturer (www.cellendes.com) before filling the microfluidic chip. The pre-gel solution mixture was filled into the abluminal side of the micro channels with the help of a syringe pump under a constant flow rate of 2.5 µL/min (Appendix A). The incubation period for complete gelation of the hydrogel was ca. 90 min.

Perfusion in the microfluidic chip during cell culture was achieved by force of gravity, employing a parallelized perfusion setup (Figure 2). To this end, the inlet and the outlet ports of each channel were fitted with reservoirs (V ≈ 1 mL). Different volumes of cell culture medium were placed into the reservoirs such that a level difference and the associated pressure difference induce flow through the channel. A parallelized peristaltic micro-pump equipped with flexible tubes reaching into both outlet and inlet reservoirs returns the perfused medium from the outlet reservoir to the inlet reservoir, thus maintaining the level difference. Pumping rate, duration of the pumping period, and duration of the waiting period as well as the level of tubing within the reservoirs can be adjusted separately as required for a particular experimental situation. Flow rate depends on the level difference between the inlet and outlet reservoir and may be adjusted between approximately 10 and 100 µL/min. Reservoirs for the perfusion set-up were made by cutting 1–5 mL pipette tips to a length of ca. 37 mm. The end of the tip was cut and slightly shortened in such a way as to fit tightly onto the inlet and outlet ports of the chip. Flexible silicone tubes (ID. 1.0 × OD. 2.0 mm) to be used with two 6-channel micro-peristaltic pumps (Takasago Fluidic) were prepared at a length of ca. 13–15 cm. The reservoirs and tubes were autoclaved prior to use. Silicone plugs were made using the SYLGARD 184 Silicone Elastomer Kit (Dow Corning Inc., Miland, MI, USA) and used to tightly cover the gel-filled channels of the MFC to prevent evaporation. The medium flowing through the perfusion channel has no contact with these plugs.

### 2.3. TEER Measurement Setup

TEER measurements employed a custom-made parallelized FFT impedance spectroscopy system used to measure impedance across cellular barriers between 1 Hz and 10 kHz (Figure 3). For the measurement, the device was taken out of the incubator and placed on the connector board. The system generates and simultaneously applies a discrete spectrum of sinusoidal signals to all culture areas on the chip in a parallel fashion. Both the voltage and current signal were recorded at high temporal resolution. Following data acquisition, Fourier transforms of both the voltage and current signals were computed and rendered the impedance spectra, which allowed for the determination of several electrical parameters of the system by using a non-linear fitting algorithm to 2-parameter (no cellular barrier present) or a 4-parameter equivalent circuit model (cellular barrier present) (Figure A2). Following the measurements, the device was remounted into the perfusion setup and placed in the incubator for further cultivation under continuous perfusion.

### 2.4. Hydrogel Preparation

The reagents were obtained from Cellendes GmbH, Reutlingen, and mixed as indicated by the manufacturer (www.cellendes.com) before the microfluidic chip was filled as shown in Table 1.

Cellendes 3-D Life Hydrogel is a two-component system consisting of a thiol-reactive polyvinyl alcohol (PVA) polymer and a thiol-functionalized crosslinker (CD-Link) which comprises a matrix-metalloproteinase cleavable peptide allowing embedded cells to cleave the crosslinker and migrate within the gel. Mixing the two components leads to the formation of stable thioether bonds and formation of the gel. Gelation time varies with pH and can be adjusted by mixing two ten-fold concentrated buffers (10× CBpH 7.2 and 10× CBpH 5.5). The lower the pH, the slower the gel formation. Peptides containing cell adhesion motifs such as RGD can be covalently attached to a fraction of the thiol-reactive groups of the thiol-reactive polymer, either prior to or after crosslinking to support adhesion of cells. Since crosslinking with CD-Link causes the pre-gel solution mixture to start to solidify in less than 3 min at room temperature, it was added to the reagent mixture just before the gel filling process was started. The prepared gel complex was placed on ice and the MFC was placed on a cold surface for gel filling to maximize the gelation time as much as possible.

### 2.5. Preparation and Seeding of Cells into Chip

HT29 Human colon adenocarcinoma cells were cultured and maintained at 5% CO_2_ at 37 °C in T75 cell culture flasks, and passaged by dissociation with 0.05% trypsin/EDTA every two to three days at a confluency of 70–90%. Cell suspension for the experiments was prepared by centrifugation of dissociated cells at 130× *g* for 7 min at RT.

A cell strainer with a pore size of 40 μm was employed to extract single HT29 cells in suspension, which is crucial to generate similarly sized tumor spheroids in the MFC. Each 30 μL volume of hydrogel composition comprised 6 μL cell suspension at a density of 2 × 10^6^ cells/mL. HT29 cells were seeded at ≈1600 cells per abluminal channel as each channel was filled with ca. 4 μL gel.

A fresh microfluidic chip was UV sterilized for 30 min. The gel was infused into the channels using a previously sterilized syringe pump set-up as described above. After gel filling, the MFC was placed in an upright position under the sterile hood and incubated for ca. 60–90 min to allow for gelation.

MDCK cells stably expressing a Vimentin specific chromobody [47] (MDCK VB6-CB) were labeled with CellTracker^TM^ Red (Invitrogen) prior to seeding and cultivation in the microchip to enable real-time visualization of the interaction of tumor spheroids (previously stained with Calcein-AM on-chip) with the MDCK cell barrier. The dye was dissolved in 20 μL high-quality DMSO to prepare a stock solution at RT; 3 μL of this stock solution was diluted in 500 μL PBS to prepare the CellTracker^TM^ Red working solution. The dye is well retained in living cells through several generations.

Each channel was filled with 10 μL of MDCK cell suspension (10^7^ cells/mL, i.e., 10^5^ cells per channel). After cell seeding, the reservoirs were covered with Parafilm and the chip was placed in an upright position in the incubator for approximately 2 h to facilitate the adhesion of cells to the gel surface. After incubation, fresh medium was added to the reservoirs and the perfusion setup was switched on. The medium was changed once every two days. The formation of a cellular barrier across the gel interface and the interaction with tumor spheroids was monitored regularly using a fluorescence microscope (Nikon Eclipse Ti).

## 3. Results

### 3.1. Chip Fabrication

Generally, devices manufactured by precision injection molding show a certain degree of shrinkage after solidification of the polymer melt in the molding tool and the subsequent cooling after demolding. In an iterative process, shadow masks used for electrode deposition were modified to reflect the actual position of the culture areas as opposed to the nominal values provided in the CAD design. Shadow masks were kept in place by magnetic forces during electrode deposition by vacuum evaporation. A precision of <50 µm was achieved for the electrode placement over the dimensions of the wellplate-sized chips. The bottom foil exhibits excellent optical properties and no autofluorescence was observed.

Priming of the MFC depended on pillar size and spacing as well as on the filling rate, V_F_. A parameter α was defined according to
(1)$$\alpha =width_{pillar}\cdot distance_{pillars}$$

Success rate decreased linearly with increasing α (Figure A3).

Perfusion rate depends on the level difference between the inlet and outlet reservoir. Flow rates ranging from 10 to 100 µL/min were applied. The resulting shear stress was calculated by multi-physics simulation (Figure A5). At a 10 µL/min flow rate, the shear stress ranges from 0.01 to 0.045 Pa, about an order of magnitude lower than what is estimated based on in vivo values [48,49].

### 3.2. Cellular Barriers in MFC

MDCK cells stably expressing a green fluorescent Vimentin chromobody (MDCK_VB6-CB) [47] adhered and formed a tight layer on both the functionalized gel interface as well as on the channel surface, i.e., chip material (Figure 4). The chip-integrated electrodes were used to determine the electrical properties of the cellular barrier by means of FFT impedance spectroscopy.

A clear correlation between TEER and the cell density of MDCK_VB6-CB cells was observed. MDCK_VB6-CB cells seeded and grown at higher densities yielded relatively dense layers, as demonstrated by TEER values of about 50–80 Ω∙cm^2^ (Figure 5B). Fitting of the data to the 4-parameter equivalent circuit (Figure A2) also yields the capacitance of the gel–medium interface (in the absence of cells) and the cellular layer and enables calculation of coverage, Θ, (Figure A2). In these preliminary experiments, a coverage of up to 100% was observed with MDCK_VB6-CB cells.

### 3.3. Tumor Spheroid Growth, Migration and Interaction with Cellular Barrier in MFC

The 3D growth of HT29 tumor cells seeded in PVA-hydrogel was studied (Figure 5C). Individualized cells were obtained by sieving the cell suspension. Cells were pelleted and mixed with the gel solution, which was then injected into the channels of the MFC. An initial growth period of 2 days without perfusion was required to observe the growth of 3D tumor spheroids. If the perfusion of medium was started immediately after the gel set, the cells would not grow in size although they remained viable as confirmed by Calcein-AM staining (not shown). The size of the spheroids increased linearly over a cultivation period of 14 days (Figure 5C) following the initial 2-day period under static conditions.

HT29 spheroids were co-cultured with a MDCK_VB6-CB cellular barrier (Figure 5A,D). MDCK_VB6-CB cells were seeded on DIV13. Spheroids remained viable as confirmed by Calcein-AM/propidium iodide staining. In case of a leaky cellular barrier, tumor spheroids were observed to remodel and leave the gel phase and grow towards the perfusion channel.

## 4. Discussion

### 4.1. Chip and Periphery Instrumentation, Workflow Compatibility

We have demonstrated a microfluidic device with integrated TEER electrodes and a complete process for establishing and analyzing co-cultures of cellular barriers with micro-tumors in a parallelized fashion.

Successful chip priming depends on the pillar width and spacing, as is to be expected as gaps between pillars act as capillary stop valves and a larger pillar-to-pillar distance will result in lower burst pressure [50].

The device fabrication relies on scalable precision injection molding technology of COC. Integrated TEER electrodes in combination with FFT-impedance spectroscopy enable rapid assessment of electrical properties of cellular barriers and coverage. However, the shrinkage of chips following demolding required iterative adaptation of shadow masks to the actual dimensions of the MFCs in order to achieve precise positioning of electrodes.

A chemically defined hydrogel was functionalized by an RGD-peptide and was shown to support prolonged growth and adhesion of MDCK cells. A cleavable peptide crosslinker (CD-link) enabled remodeling by 3D tumor spheroids and their outgrowth into the perfusion channel.

The incubator-compatible perfusion system enables gravitation-driven unidirectional flow vs. bidirectional flow as found in other models [51]. It integrates seamlessly into common cell culture workflows.

However, the need for certain improvements also became apparent: in a future design, fluid connectors could be implemented in a standard industry format and spacing on the chip. Silicone tubing used with peristaltic pumps may cause unwanted absorption of compounds [41,42] and it also contributes to the overall dead volume in the system, which should be minimized. Accordingly, there is a need for parallelized, low dead volume pumping schemes to support the application of parallelized organ-on-chip in vitro models. Finally, a third channel could be added to the cell culture areas to enable assessment of substance transfer and retrieval from the in vitro model for further analysis [52].

### 4.2. Cell Culture in MFC

The feasibility and versatility of cell culture in the MFC in general was demonstrated by cultivation of the cellular barriers of MDCK, and also in the co-culture with HT29 tumor spheroids in the adjacent gel phase.

TEER measured using the integrated electrodes generally was significantly lower than that seen in earlier reports [53,54,55]. We hypothesize that the interface between cellular layer and chip material may represent an ion leakage in cases where cells only adhere on the functionalized gel surface. In contrast to designs with a co-planar orientation of the cellular barrier [2,56,57] with the chip plane, in our design, the cellular barrier is composed of 22 individual patches with the outer rim contacting the chip material. The resistance is very sensitive towards even minor ion leaks [2,58] (Figure A3). On the other hand, capacitance provides a reliable measure of overall coverage (Figure A2).

The inhibition of cell growth in HT29 tumor spheroids under perfusion at the beginning of the culture may be due to the unintended dilution of factors secreted by the cells in case of an excess liquid to cell ratio and highlights the issue of finding and establishing the proper liquid to cell ratio (LoC) in microfluidic in vitro models [59]. As relatively high LoC values are often found in microfluidic in vitro models, this should receive more attention in future technical developments of organ-on-chip (OoC) models. However, proper manipulation of minute volumes of fluid while maintaining constant perfusion and avoiding effects such as pinning of meniscus to edges and evaporation certainly poses a significant technical challenge.

## 5. Conclusion

In summary, we demonstrated a parallelized microfluidic device and a comprehensive workflow to study and analyze co-cultures of cellular barriers and 3D tumor spheroids under continuous unidirectional flow.

## Figures and Tables

**Figure 1 biosensors-11-00314-f001:**
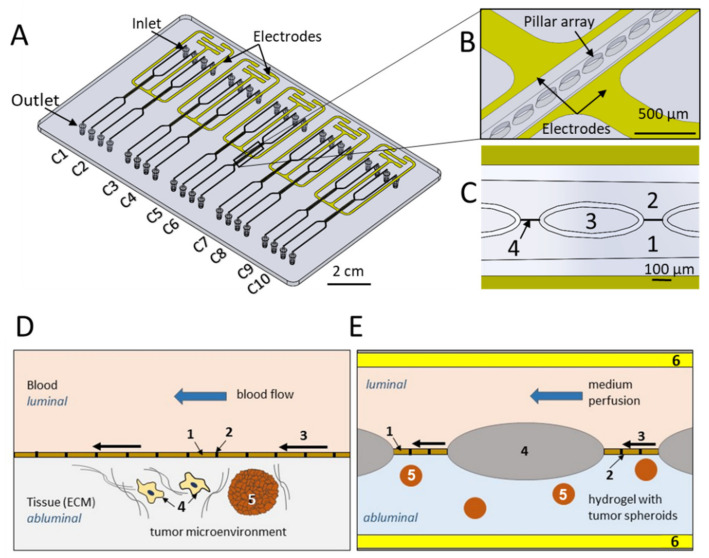
Chip technology: (**A**) precision injection molded device with multi-well-plate footprint, comprising 10 culture areas C1 to C10 with two channels, inlets and outlets per culture area. (**B**) Electrodes line the culture area with contact pads positioned at the plate edge (**A**). (**C**) Within the culture areas, both channels are separated by an array of pillars. Aqueous, low viscous hydrogel solution with or without cells may be introduced in one of the channels (“abluminal channel”) whereby gaps between pillars act as capillary stop valves preventing spilling of the gel solution into the adjacent channel (“perfusion channel” or “luminal channel”). (**D**) Scheme of tumor–vascular interface showing endothelial barrier (1) with cells forming tight junctions (2). Blood flow results in shear forces (3). Immune cells (4) invade tissue to attack tumor tissue (5). (**E**) Scheme of chip according to this study mimicking the tumor–vascular micro-environment by co-culture of tumor spheroids (5) in a hydrogel matrix with a cellular barrier (1) grown on the interface formed between pillars (4). Electrodes (6) located at the periphery of both channels enable measurement of the electrical impedance across the interface.2.3. Perfusion setup.

**Figure 2 biosensors-11-00314-f002:**
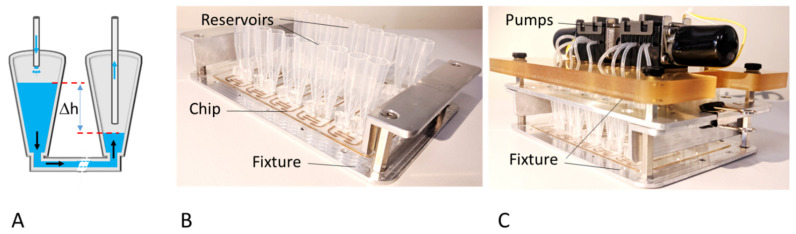
Parallelized perfusion setup: (**A**) Scheme of gravitation-driven flow. Medium reservoirs (V ≈ 1 mL) were placed on the channel inlet and outlet. Tubes reaching into the reservoirs in connection with micro-peristaltic pumps that operated intermittently were used to move perfused medium from the outlet reservoir back to the inlet reservoir, thus maintaining the level difference, Δh, driving unidirectional medium flow. (**B**) Chip in fixture with medium reservoirs attached on fluid ports of MFC. (**C**) Perfusion setup with 10 channel microfluidic pumps mounted on top of the chip fixture to yield an easy-to-use and incubator compatible perfusion system.

**Figure 3 biosensors-11-00314-f003:**
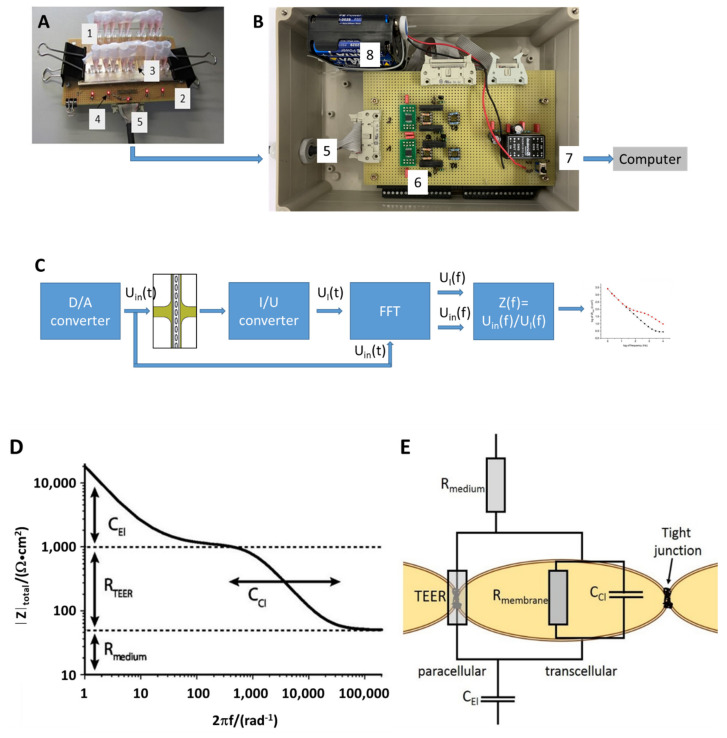
TEER measurement setup: (**A**) Chip (1) with medium-filled reservoirs (3) mounted on electrical interface fixture (2). LEDs (4) indicate proper connection of chip contacts to the measurement system. A cable (5) delivers the voltage signal, U_in_(t), to the 10 culture areas simultaneously and feeds individual current signals, U_I_(t), to the current-to-voltage converter (6) in the measurement system. (**B**) Measurement system comprising a power supply (8), a digital/analog converter (D/A), current-to-voltage converters (I/U), digital switches, and data acquisition board (National Instruments USB9012) connected to a computer via USB-cable (7). (**C**) Schematic circuit diagram: the D/A-converter provides a voltage signal synthesized from a set of discrete frequencies to the chip. Voltage amplitudes may be adjusted by the LabView application software. Fourier transform of both current and voltage signals yields frequency spectra thereof and allows for calculation of the impedance spectra Z(f) of 10 channels within seconds. (**D**) Impedance spectra measured in the MFC exhibit distinct features reflecting the electrical properties (adapted from [46] with permission) (**E**) of electrodes, C_El_, medium, R_medium_, and interface with or without cellular barrier, TEER, R_membrane_ and C_dl_.

**Figure 4 biosensors-11-00314-f004:**
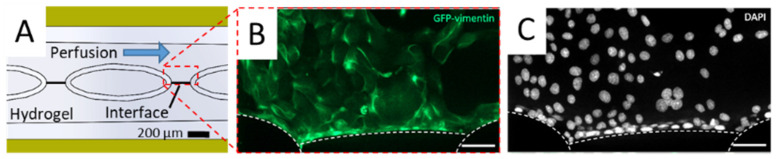
Fluorescence microscopy characterization of cellular barriers in MFC. (**A**) Schematic figure of location of cellular barrier on the interface between gel phase and perfusion channel. (**B**), (**C**) Cellular barrier comprised of MDCK_VB6-CB cells. Cells are located on the interface between the perfusion channel and hydrogel as well as on the surface of micro pillars and proliferate to eventually cover the complete surface of the perfusion channel. Cells expressing green fluorescent Vimentin chromobody (VB6-CB B) were fixed with PFA and DAPI staining was performed (**C**). Cells were seeded at a density of 10^5^ cells per channel at passage 3. Staining was performed at day five after seeding. Single slice confocal microscope image at bottom of the channel. Scale bar 50 µm.

**Figure 5 biosensors-11-00314-f005:**
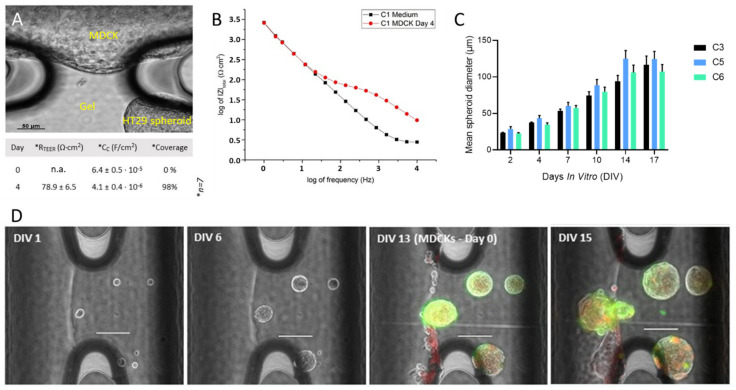
Co-culture of HT29 tumor spheroids with a cellular barrier of MDCK_VB6-CB cells. (**A**) Typical micrograph of culture area no. 1 showing the perfusion channel and gel interface between pillars completely lined by MDCK_VB6-CB cells and HT29 tumor spheroid growing in gel phase. An average TEER of 78.9 +/− 6.5 Ωcm^2^ at DIV4 was observed from *n* = 7 culture areas. The interfacial capacitance, C_c_, yielded an average coverage of 98%. (**B**) Typical impedance spectra of interface (culture area no. 1, corresponding to image in (**A**) without a cellular barrier (black dots) at day 0 and at day 4 of MDCK_VB6-CB cells (red dots), respectively. TEER: 58.7 Ωcm^2^; lines represent the least square fit to data using the equivalent circuit display in Figure A2A. (**C**) Growth of HT29 tumor spheroids in three different channels during a period of 17 days, with 2 days under static conditions, and thereafter under continuous perfusion. Mean diameter of spheroids increased from 25 µm to 120 µm. Error bars indicate standard deviation calculated from n = 8 micro-spheroids. (**D**) Micrographs showing the growth of tumor spheroids in the gel phase under continuous perfusion, their interaction with a leaky cellular barrier of MDCK_VB6-CB cells (also labelled with CellTracker^TM^ Red) and finally the outgrowth of a tumor spheroid into the perfusion channel at day 15.

**Table 1 biosensors-11-00314-t001:** Reagent volumes for 120 μL gel for one MFC.

3-D Life Reagents	V (μL)	V (μL)
10× CB (80% pH 7.2, 20% pH 5.5)	2.4	9.6
Water	15.3	61.2
SG-PVA (30 mmol/L SH-reactive groups)	2.5	10.0
RGD Peptide (20 mmol/L SH groups)	0.8	3.2
Cell suspension (HT29)	6.0	24.0
CD-Link (20 mmol/L SH groups)	3.0	12.0
Total	30.0	120.0

## Data Availability

Not applicable.

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
