# Peer review of "Parallelizable Microfluidic Platform to Model and Assess In Vitro Cellular Barriers: Technology and Application to Study the Interaction of 3D Tumor Spheroids with Cellular Barriers"

_biosensors, 2021, doi:10.3390/bios11090314_

Round 1

Reviewer 1 Report

Summary: The paper describes a parallelized microfluidic device in a multi-well format with ten cell culture chambers to demonstrate cellular barriers. The technology has been demonstrated with co-culture conditions of cellular barrier models with 3D tumor spheroids. The integrated chip with an FT-impedance measurement system enabled transepithelial or transendothelial electrical resistance (TEER) measurements and assess barrier tightness. The technology although was manufactured using microinjection materials has several materials and methods that operate with low throughput components and non-ideal consumables. The author has discussed potential steps to be adopted in the future to improve the existing technology and encourage its adoption in the future.

Comments:

  1. The group has correctly indicated the existing limitation of PDMS chips, namely, “PDMS-based chips tend to adsorb substances”. With this knowledge, the authors have correctly taken the necessary step to design their devices in COC material. The plastic bottom can cause background fluorescence and need to be discussed in the current device. The imaging limitation must be added to the discussion.
  2. What flow rates are achieved with the gravity-driven forces? Include the flow rates under different conditions and how much shear forces are generated under this flow? How does the shear compare with the physiological conditions?
  3. Add more explanation for the data presented in the paper. For example, if the author presented standard deviation or standard error of the mean in Figure 5(c). What statistical test was used? What is represented as (*) in the figure?

Minor edits:

  1. Line 8-13: provide full address.
  2. Line Line 385: introduce OoC.
  3. Figure 3 (E): the figure legend does not introduce Rmembrane.

Author Response

Reviewer I:

Summary: The paper describes a parallelized microfluidic device in a multi-well format with ten cell culture chambers to demonstrate cellular barriers. The technology has been demonstrated with co-culture conditions of cellular barrier models with 3D tumor spheroids. The integrated chip with an FT-impedance measurement system enabled transepithelial or transendothelial electrical resistance (TEER) measurements and assess barrier tightness. The technology although was manufactured using microinjection materials has several materials and methods that operate with low throughput components and non-ideal consumables. The author has discussed potential steps to be adopted in the future to improve the existing technology and encourage its adoption in the future.

We thank the reviewers for their valuable feedback and address their comments as outlined in detail below.

Comments:

    The group has correctly indicated the existing limitation of PDMS chips, namely, “PDMS-based chips tend to adsorb substances”. With this knowledge, the authors have correctly taken the necessary step to design their devices in COC material. The plastic bottom can cause background fluorescence and need to be discussed in the current device. The imaging limitation must be added to the discussion.

Autofluorescence of the bottom cover made of COC has been negligible in our chip and did not hamper analysis of the cell culture in any way.

    What flow rates are achieved with the gravity-driven forces? Include the flow rates under different conditions and how much shear forces are generated under this flow? How does the shear compare with the physiological conditions?

The range of achievable flowrates was added to paragraphs 2.3 and 3.1, respectively. A graph (Figure A5) was added showing the result of a multiphysics simulation of the shear stress at a flow rate of 10µl/min. Also, literature references of in vivo data were added in paragraph 3.1.

    Add more explanation for the data presented in the paper. For example, if the author presented standard deviation or standard error of the mean in Figure 5(c). What statistical test was used? What is represented as (*) in the figure?

Data in graph 5C are displayed as mean of n=8 spheroids with the error bars indicating standard deviation. A statement was added to the figure caption.

We reviewed and amended the figure captions carefully.

Minor edits:

    Line 8-13: provide full address.

amended

    Line Line 385: introduce OoC.

Amended

    Figure 3 (E): the figure legend does not introduce Rmembrane.

Amended.

Reviewer 2 Report

Stelzle et al. described a microfluidic device which was fabricated by microinjection molding rather than commonly used PDMS. By integrating a FT-impedance measurement system, this device is able to measure the TEER value of the epithelial cells cultured in the chip in real time. The authors used MDCK cells as the model, the author has successfully demonstrated the power of this semi-automatic organ-on-a-chip device. However, there are several issues I want the authors to address before consider accepting:

  1. Reference 15-19, 20 should be 15-20.
  2. The cited papers such as 15-17 are too outdated, could you update with several newer references?
  3. From line 66-79, it lacks references to support author’s opinion, maybe check a nature review paper as an example:  Organs-on-chips: into the next decade, Nature Review Drug Discovery, 2021.
  4. For reference 21, I would suggest the author cite the more recent review: “Development of Polymeric Nanoparticles for Blood–Brain Barrier Transfer—Strategies and Challenges”
  5. The author used another material and fabrication method other than PDMS-based lithography, may I know does the chip have nice light transparency? I cannot tell from the figures.
  6. The Figure 4 the author claimed immunostaining/immunocytochemical staining which is wrong! There was no antibody used in the process. The fluorescence came from the GFP of the cells. Therefore, I would like to see the real immunofluorescent staining of the tight junction/adherent junction proteins, such as ZO-1 or VE-Cadherin or CD31 of the chip. The author should change the wrong terms throughout the manuscript as well.
  7. For the last experiment, has the author compared the TEER with and without the tumor sphere? This is important.
  8. Figure 5D, why the MDCK cells did not show any fluorescence?

Author Response

Reviewer II:

Stelzle et al. described a microfluidic device which was fabricated by microinjection molding rather than commonly used PDMS. By integrating a FT-impedance measurement system, this device is able to measure the TEER value of the epithelial cells cultured in the chip in real time. The authors used MDCK cells as the model, the author has successfully demonstrated the power of this semi-automatic organ-on-a-chip device. However, there are several issues I want the authors to address before consider accepting:

We appreciate the reviewer’s comments and address them as outlined in detail below.

    Reference 15-19, 20 should be 15-20.

amended

    The cited papers such as 15-17 are too outdated, could you update with several newer references?

Additional more recent references were added.

    From line 66-79, it lacks references to support author’s opinion, maybe check a nature review paper as an example:  Organs-on-chips: into the next decade, Nature Review Drug Discovery, 2021.

Edited and amended, selected references added.

    For reference 21, I would suggest the author cite the more recent review: “Development of Polymeric Nanoparticles for Blood–Brain Barrier Transfer—Strategies and Challenges”

We thank the reviewer for drawing our attention to this very relevant review, which we included in the citation.

    The author used another material and fabrication method other than PDMS-based lithography, may I know does the chip have nice light transparency? I cannot tell from the figures.

In paragraph 2.1 we explicitly state now, that COC is transparent and exhibits low autofluorescence.

    The Figure 4 the author claimed immunostaining/immunocytochemical staining which is wrong! There was no antibody used in the process. The fluorescence came from the GFP of the cells. Therefore, I would like to see the real immunofluorescent staining of the tight junction/adherent junction proteins, such as ZO-1 or VE-Cadherin or CD31 of the chip. The author should change the wrong terms throughout the manuscript as well.

Amended.

The focus of this paper is on the features of the chip and the use thereof. We certainly agree, that in future experiments with a focus on biological function of cellular barriers, the mentioned markers should be applied.

    For the last experiment, has the author compared the TEER with and without the tumor sphere? This is important.

As can be seen in Fig.5A, MDCK were co-cultured with tumor spheroids and values of RTEER obtained thereof in case of tight cellular barriers are reported (average of n=7). In Fig. 5B, representative impedance spectra obtained with and without MDCK cell layer, respectively, are shown. In the case of Fig.5D, the cellular barrier was leaky and impedance would be low (comparable to the black curve displayed in Fig.5B).

We show, that measurement of TEER is possible both in the presence and absence of the cellular barrier and/or tumor spheroids. This capability will be exploited in future research to possibly use TEER to monitor the interaction of tumor spheroids with an intact cellular barrier.

    Figure 5D, why the MDCK cells did not show any fluorescence?

The fluorescence intensity of the vimentin chromobody was weak in comparison to the calcein signal of the – in comparison massive - tumor spheroids. However, the additional (red) staining of the MDCK cells using CellTrackerTM Red is clearly visible in the cells at the gel / channel interface.

Reviewer 3 Report

The manuscript entilted "Parallelizable microfluidic platform to model and assess in vitro cellular barriers: technology and application to study the interaction of 3D tumor spheroids with cellular barriers" presents lab-on-chip platform dedicated to investigate development of 3D spheroid culture and its interaction with cellular MDCK microenvironment with simultaneous possibility of FT-impedance measurements. The manuscript raised significant and very recent literature trend covering the subject of microfluidic-based tumor cultivation in the conditions thoroughly imitating in vivo environment, to ultimately match, e.g. the best and personalized anti-cancer therapy. Although the scope is vital, and the results promising, few aspects have to be discussed in detail before the manuscript can be published.

Major comments:

1) The Authors state (line 85) that “A number of research groups have been successful in developing models of cellular barriers with important insights into cell culture conditions, morphology, viability and integrity of barriers in a microfluidic chip and their potential in drug screening applications, particularly in the context of modelling the blood-brain barrier [18, 19, 26-29]. Also, devices to model aspects of cancer physiology and development have been presented [30, 31]. Most of these devices, however, were fabricated from PDMS. While this presents the advantages of ease of fabrication, PDMS-based chips tend to adsorb substances [32, 33] (…)”.

It is truth but the Authors should additionally mention that COC polymer material, which the LOC was fabricated out of, has also some limitations, related to its hydrophobic surface and spontaneous adsorption of some molecules as well (Biomicrofluidics. 2012 Mar; 6(1): 012822–012822-12). The Authors should rather underline the fabrication simplicity of COC-based LOCs, and soften the description on PDMS biocompatibility, especially taking into consideration its further application as silicone plugs in this paper. In general, I feel a lack of COC material description in the Introduction part including both its advantages and disadvantages (what about time degradation?), matching substantial references.

2) The Authors should also supplement Introduction part with additional paragraph referring to bioprinting technique utilizing hydrogel inks, which is an emerging field now. Lot of articles can be found on its application for 3D spheroid platforms. Please, compare your work and underline novelty aspects with regard to technology.

Minor comments:

1) The title of the article is quite long, think about the shorter one, e.g. "Parallelizable microfluidic platform to model and assess the interaction of 3D tumor spheroids with cellular barriers".

2) CNS – please define the abbreviation (line 43).

3) What were the exact bonding parameters (line 128-129)? Did you provide additional surface modification to enhance optical/mechanical features and minimize the adsorption effects?

4) The Authors write that the cells were introduced to the channel later (line 139). It is not compatible with line 220, please explain.

5) How did you organize the observation of cells? Was it real time? The PCB from the bottom and large tips from above prevent to obtain good visibility.

6) Please explain the RT acronym.

7) What was the height of the pillars? Some cross-section view should be added, maybe SEM photographs to see the details of the structure.

8) The cultivation time was 14 or 17 days? 17 days cover the time for cell adhesion? (Line 319, 334).

9) It is not clear enough to each channel was introduced MDCK cell suspension. In line 220 there is an information that to each. The cells adhere only selectively to the gel surface? What about the other parts of the chip? I suggest the improvement of the Fig. 1 to clearly indicate the presence of MDCK cells.

10) What was the volume of self-made medium reservoirs? Was it enough for your 14-days culture, or you had to complement it during the experiment?

11) Did you conduct any flow analysis utilizing Comsol/Ansys before you proceed to the LOC design? How did you choose the pillar geometry to get the effect of stop valves?

12) I suggest to extract Conclusion part (line 389-391) or remove the last paragraph.

Author Response

Reviewer III:

The manuscript entilted "Parallelizable microfluidic platform to model and assess in vitro cellular barriers: technology and application to study the interaction of 3D tumor spheroids with cellular barriers" presents lab-on-chip platform dedicated to investigate development of 3D spheroid culture and its interaction with cellular MDCK microenvironment with simultaneous possibility of FT-impedance measurements. The manuscript raised significant and very recent literature trend covering the subject of microfluidic-based tumor cultivation in the conditions thoroughly imitating in vivo environment, to ultimately match, e.g. the best and personalized anti-cancer therapy. Although the scope is vital, and the results promising, few aspects have to be discussed in detail before the manuscript can be published.

We appreciate the reviewer’s comments that helped us in clarifying several important aspects of our paper and address them as outlined in detail below.

Major comments:

1) The Authors state (line 85) that “A number of research groups have been successful in developing models of cellular barriers with important insights into cell culture conditions, morphology, viability and integrity of barriers in a microfluidic chip and their potential in drug screening applications, particularly in the context of modelling the blood-brain barrier [18, 19, 26-29]. Also, devices to model aspects of cancer physiology and development have been presented [30, 31]. Most of these devices, however, were fabricated from PDMS. While this presents the advantages of ease of fabrication, PDMS-based chips tend to adsorb substances [32, 33] (…)”.

It is truth but the Authors should additionally mention that COC polymer material, which the LOC was fabricated out of, has also some limitations, related to its hydrophobic surface and spontaneous adsorption of some molecules as well (Biomicrofluidics. 2012 Mar; 6(1): 012822–012822-12). The Authors should rather underline the fabrication simplicity of COC-based LOCs, and soften the description on PDMS biocompatibility, especially taking into consideration its further application as silicone plugs in this paper. In general, I feel a lack of COC material description in the Introduction part including both its advantages and disadvantages (what about time degradation?), matching substantial references.

Following the reviewer’s suggestion, we added a statement regarding the hydrophobicity of COC and the possibility of surface treatment and added a reference to a preferred method.

Nevertheless, PDMS in comparison exhibits much larger porosity and certainly is known (as cited) as quite problematic with respect to substance absorption.

The silicone plugs do not connect to the perfusion channel but are applied to prevent evaporation of fluid from the gel phase. Therefore, substance adhesion to these plugs cannot occur. This is now explicitly mentioned in the text.

2) The Authors should also supplement Introduction part with additional paragraph referring to bioprinting technique utilizing hydrogel inks, which is an emerging field now. Lot of articles can be found on its application for 3D spheroid platforms. Please, compare your work and underline novelty aspects with regard to technology.

We agree with the reviewer that there are significant and valuable efforts in utilizing 3D-bioprinting to generate microtissues. We feel, however, that a truly useful discussion of these reports is beyond the scope of our paper, as we are reporting on microspheroids grown from single cells embedded in a gel matrix within a microfluidic device.

Minor comments:

1) The title of the article is quite long, think about the shorter one, e.g. "Parallelizable microfluidic platform to model and assess the interaction of 3D tumor spheroids with cellular barriers".

In our paper, we present this technology platform and then go on to report on a specific application as an example of how this platform may be used. Yet many other applications are quite as well conceivable using this technology. We therefore prefer to keep the title as is.

2) CNS – please define the abbreviation (line 43).

Amended

3) What were the exact bonding parameters (line 128-129)? Did you provide additional surface modification to enhance optical/mechanical features and minimize the adsorption effects?

The bonding parameters are proprietary and unfortunately cannot be disclosed.

No special treatment was applied to the bonding foil. There is therefore nothing we could add to the description of the bonding process.

4) The Authors write that the cells were introduced to the channel later (line 139). It is not compatible with line 220, please explain.

We seed tumor cells by mixing the hydrogel solution with a cell suspension and fill the abluminal channel with this mixture. After setting of the hydrogel, the cell suspension comprising epithelial or endothelial cells intended to form the cell barrier layer are filled into the luminal channel where they adhere to the interface between gel phase and luminal channel. We added a few words to both paragraphs to make the difference clear.

5) How did you organize the observation of cells? Was it real time? The PCB from the bottom and large tips from above prevent to obtain good visibility.

At this point, impedance measurement was performed discontinuously by taking the device out of the incubator and the perfusion setup and placing it onto the connector board. After completion of the measurement, the device was again mounted in the perfusion setup and placed in the incubator for further cultivation. This is now explicitly described in paragraph 2.4. Microscopic inspection from below is feasible with the device being mounted in the perfusion setup using an inverted microscope.

6) Please explain the RT acronym.

Amended.

7) What was the height of the pillars? Some cross-section view should be added, maybe SEM photographs to see the details of the structure.

The channel depth (170µm) was already included in table A1, but was also added to both paragraph 2.1 as well as to the caption of Figure A1.

8) The cultivation time was 14 or 17 days? 17 days cover the time for cell adhesion? (Line 319, 334).

The text in both figure caption and paragraph 3.3 was amended to state that there was an initial 2 day period of cultivation under static conditions prior to the onset of perfusion.

9) It is not clear enough to each channel was introduced MDCK cell suspension. In line 220 there is an information that to each. The cells adhere only selectively to the gel surface? What about the other parts of the chip? I suggest the improvement of the Fig. 1 to clearly indicate the presence of MDCK cells.

As can be seen from Fig.4, adhesion of MDCK cells occurred throughout the channel not exclusively on the gel interface. A clarifying statement was added to Figure caption 4.

10) What was the volume of self-made medium reservoirs? Was it enough for your 14-days culture, or you had to complement it during the experiment?

The volume was 1ml; Figure caption 2 was amended.

As is described in the caption of figure 2, the medium was continuously re-perfused by means of the peristaltic pumps mounted on top of the chip fixture.  The medium was changed every other day as stated in paragraph 2.6.

11) Did you conduct any flow analysis utilizing Comsol/Ansys before you proceed to the LOC design? How did you choose the pillar geometry to get the effect of stop valves?

Multiphysics simulations of electrical fields and hydrodynamic forces were conducted in order to determine flow and shear forces. See Figure A5 added in the revised manuscript.

12) I suggest to extract Conclusion part (line 389-391) or remove the last paragraph.

Following the reviewer's suggestion a section “5. Conclusion” was added.

Reviewer 4 Report

The authors present a microfluidic culture chamber to model cell barriers. The chip contains an integrated electrode and corresponding hardware was designed to measure “barrier” impedance. While the authors have successfully designed and built a microfluidic solution for modeling the interface between tumor spheroids embedded in gel and a continuous flow, the application of this technology was extremely limited, and the parallelization under-utilized. It would have been wonderful to see a more thorough study that used this technique to uncover something of biological interest. In conclusion, this paper presents a new technology, but falls short of a real biological experiment. The figures are rather poor in quality and presentation.

Specific comments

The authors state that TEER measurements allow them to “asses real-time barrier tightness.” However, there is not a true cell barrier in their device.  Cells embedded in gel on one side and cells flowing on the other. Can this be claimed to be a true barrier like the BBB?

The authors should more clearly explain why impedance is a good/important way to characterize barrier properties. How can impedance measurements be used to say anything particularly interesting about the biology of tumor growth or other interfaces? Is there any empirical support for the idea that this measurement can characterize leakiness or permeability of the barrier? Are control experiments not possible that could relate barrier properties to the impedance measurements?

It would have been interesting to see an intervention of some type whereby a drug or perturbation was introduced, and the properties of the interface were measured. This was part of the motivation for the development of this chip, right? Although I applaud the engineering and bioMEMS aspects of this paper, there is no biological rigor. Indeed, the one figure with data collected from a biological experiment (Fig 5B) had no error bars.

Partnership with a lab that has interesting biological question (and controls) related to cellular barriers would allow this technology to be better tested and applied to an interesting biological question. 

Author Response

Reviewer IV:

The authors present a microfluidic culture chamber to model cell barriers. The chip contains an integrated electrode and corresponding hardware was designed to measure “barrier” impedance. While the authors have successfully designed and built a microfluidic solution for modeling the interface between tumor spheroids embedded in gel and a continuous flow, the application of this technology was extremely limited, and the parallelization under-utilized. It would have been wonderful to see a more thorough study that used this technique to uncover something of biological interest. In conclusion, this paper presents a new technology, but falls short of a real biological experiment. The figures are rather poor in quality and presentation.

We greatly appreciated the reviewer’s comments, which helped us to improve several aspects of our paper. We addressed the comments as outlined in detail below.

Specific comments

The authors state that TEER measurements allow them to “asses real-time barrier tightness.” However, there is not a true cell barrier in their device.  Cells embedded in gel on one side and cells flowing on the other. Can this be claimed to be a true barrier like the BBB?

In Fig. 5A, B, a cellular barrier of MDCK cells is shown along with impedance measurements of the interface between the gel and the perfusion channel without and with cellular barrier, respectively. In case of the presence of a cellular barrier, the impedance spectrum shows a pronounced increase in the frequency range above approx. 50Hz, which is indicative for increased resistance according to the evaluation using the equivalent circuit (Figure A2). This clearly demonstrates our claim of a cellular barrier present in the device. Also, Fig. 4B, C clearly show cells adhering at the interface between the gel phase (indicated by a white dotted line) and the perfusion channel.

The authors should more clearly explain why impedance is a good/important way to characterize barrier properties. How can impedance measurements be used to say anything particularly interesting about the biology of tumor growth or other interfaces? Is there any empirical support for the idea that this measurement can characterize leakiness or permeability of the barrier? Are control experiments not possible that could relate barrier properties to the impedance measurements?

Impedance measurement is an established method to assess the integrity and tightness of cellular barriers (see also references cited in the paper). We demonstrate the measurement of electrical properties of the interface between gel phase and perfusion channel with or without cellular barrier by means of electrodes integrated in the chip and Fourier-Transform impedance spectroscopy.

Resistance indeed is very sensitive to even very small defects of the cellular barrier. This is now explained in more detail in Figure A3 (annex).

Following the reviewer’s suggestion to present a control experiment, an additional figure (now Figure A3 C) was added to the annex, showing diffusion of FITC dextran, MW 4kD, with and without cellular barrier, respectively, again clearly demonstrating the effect of a cellular barrier.

It would have been interesting to see an intervention of some type whereby a drug or perturbation was introduced, and the properties of the interface were measured. This was part of the motivation for the development of this chip, right? Although I applaud the engineering and bioMEMS aspects of this paper, there is no biological rigor. Indeed, the one figure with data collected from a biological experiment (Fig 5B) had no error bars.

We certainly agree with the reviewer that this system should be evaluated by testing drug action on microtumors with or without cellular barriers and this is exactly what this system will be used for in the future.

However, the objective of the present manuscript was to present the technological prerequisite for this type of experiments by devising a parallelized microfluidic culture system, suitable periphery instrumentation and measurement technology as well as the comprehensive workflow for using it. The exemplary biological data presented was intended to demonstrate i) parallelized culture and analysis of cellular barriers and ii) feasibility of coculture of cellular barriers with tumor microspheroids under continuous perfusion, thereby demonstrating the wealth of potential applications of this technology.

With regard to the graphs displayed in Figure 5B: the dots represent raw data of the impedance spectra, the lines are least square fits using the four-parameter equivalent circuit model. The excellent agreement between measured data and fitting curve shows how precise these data are. A statement was added to the figure caption.

Partnership with a lab that has interesting biological question (and controls) related to cellular barriers would allow this technology to be better tested and applied to an interesting biological question.

The scientific questions are stated in the introduction and will be addressed using this device in future research. In addition, we certainly welcome any suggestions in this regard.

Round 2

Reviewer 3 Report

The Authors have addressed the major comments contained in the review. Although I feel that a few words in the Introduction part that would broader describe the current technological approaches concerning hydrogel microculture could be written (e.g. bio-print), I accept the manuscript in this form and truly recommend the publication.

Reviewer 4 Report

With regard to the graphs displayed in Figure 5B: the dots represent raw data of the impedance spectra, the lines are least square fits using the four-parameter equivalent circuit model. The excellent agreement between measured data and fitting curve shows how precise these data are. A statement was added to the figure caption.

It is not possible to estimate precision with a single measurement. Precision refers to how reproducible your measurement is. The fit tells you how well your data fits a model. My question was: if the authors have successfully parallelized a system for studying 10 barriers at a time, shouldn’t they be able to assess (and present) the reproducibility of their system?  

The scientific questions are stated in the introduction and will be addressed using this device in future research. In addition, we certainly welcome any suggestions in this regard.

1) Create barriers with different properties and characterize them - are your results consistent with what you'd expect of these different barriers? 2) Use control drugs with known cytotoxic or barrier-modifying properties to modulate barrier function - does your system faithfully detect these changes? 3) Screen for drugs that modify barrier permeability in an interesting biological way.